# Contribution of Dysregulated DNA Methylation to Autoimmunity

**DOI:** 10.3390/ijms222111892

**Published:** 2021-11-02

**Authors:** Samanta C. Funes, Ayleen Fernández-Fierro, Diego Rebolledo-Zelada, Juan P. Mackern-Oberti, Alexis M. Kalergis

**Affiliations:** 1Instituto Multidisciplinario de Investigaciones Biológicas-San Luis (IMIBIO-SL), Consejo Nacional de Investigaciones Científicas y Técnicas(CONICET), Universidad Nacional de San Luis (UNSL), San Luis CP 5700, Argentina; samanta.funes@gmail.com; 2Millennium Institute on Immunology and Immunotherapy, Departamento de Genética Molecular y Microbiología, Facultad de Ciencias Biológicas, Pontificia Universidad Católica de Chile, Santiago 8320000, Chile; alfernandez1@uc.cl (A.F.-F.); dmrebolledo@uc.cl (D.R.-Z.); 3Instituto de Medicina y Biología Experimental de Cuyo—IMBECU-CONICET, Facultad de Ciencias Médicas, Universidad Nacional de Cuyo, Mendoza CP 5500, Argentina; jpmackern@mendoza-conicet.gob.ar; 4Departamento de Endocrinología, Facultad de Medicina, Pontificia Universidad Católica de Chile, Santiago 8320000, Chile

**Keywords:** DNA methylation, epigenetic, systemic autoimmunity, rheumatoid arthritis, CpG

## Abstract

Epigenetic mechanisms, such as DNA methylation, histone modifications, and non-coding RNAs are known regulators of gene expression and genomic stability in cell growth, development, and differentiation. Because epigenetic mechanisms can regulate several immune system elements, epigenetic alterations have been found in several autoimmune diseases. The purpose of this review is to discuss the epigenetic modifications, mainly DNA methylation, involved in autoimmune diseases in which T cells play a significant role. For example, Rheumatoid Arthritis and Systemic Lupus Erythematosus display differential gene methylation, mostly hypomethylated 5′-C-phosphate-G-3′ (CpG) sites that may associate with disease activity. However, a clear association between DNA methylation, gene expression, and disease pathogenesis must be demonstrated. A better understanding of the impact of epigenetic modifications on the onset of autoimmunity will contribute to the design of novel therapeutic approaches for these diseases.

## 1. Introduction

Epigenetics constitutes the study of molecular modifications that alter genomic function without changing the DNA sequence [1]. The term was coined in the 1940s, referring to the interaction of genes with their products (proteins) and their effect on phenotype [2]. Since then, much progress has been made in understanding epigenetic regulatory mechanisms and how epigenetic changes can become crucial in disease onset and progression [3].

Autoimmune diseases are characterized by the breakdown of self-tolerance and the presence of self-reactive immune cells [4]. Among them, we will focus on the most frequent diseases, including Rheumatoid Arthritis (RA), Systemic Lupus Erythematosus (SLE), and others, such as Multiple Sclerosis (MS), Sjogren’s Syndrome (SS), and Psoriasis. Although the etiology of autoimmune diseases is associated with a complex genetic susceptibility, it is clear that genes are not the only factors contributing to disease [5]. Indeed, when evaluating the development of autoimmune diseases in genetically identical monozygotic twins, environmental factors can contribute substantially to developing autoimmune disorders [6]. Epigenetic mechanisms can be influenced by environmental factors and be heritable, including microRNAs, post-transcriptional modifications (PTMs) of histones, and DNA methylation [7]. These mechanisms alter chromatin architecture, control the accessibility to transcriptional regulatory factors, and regulate gene transcription rates.

Many immune cells showed reduced DNA methylation among pro-inflammatory genes during autoimmune diseases, which may be linked to gene expression induction [8]. However, there is much to be understood about the role of epigenetic modifications in autoimmune diseases. Identifying master epigenetic changes will improve their use as biomarkers for epigenetic risk, contributing to therapies based on modifying the epigenetic signature.

### 1.1. Overview of DNA Methylation in Immune Cells

The cytosine methylation in a 5-prime cytosine-guanine dinucleotide CCGG site (CpG) inside a gene locus is linked to gene repression [9]. However, the whole genome methylation is much more complex. There are regions called CpG islands (CGIs) in the genome that are sequences enriched with at least 60% CpG arrangements mainly located in the gene promoter sequence, often grouped in clusters, and found near the transcription start sites [10]. Most of the CpG dinucleotide sites in the genome are methylated [9]. Methylation of a specific CpG site is affected and modulated by the methylated status of neighboring CpG sites [11]. Non-promoter CpG sites, including gene body (gene exons and introns) and sequences located at different distances from the transcription start sites called CpG shores, shelves, and open sea regions, could also be methylated [10]. CpG shores and CpG shelves are clustered within 2 kb and between 2 to 4 kb from promoter-linked CGIs. These last CpG regions are linked to differential cell-specific gene expression [11]. Understanding the role of methylation on these non-promoter CpG sites, such as open sea regions, is much more complex than CGIs, and some diseases displayed differential methylation in these regions [11,12]. Methylated, hemimethylated, and unmethylated CpG sites could recruit DNA methyltransferases (DNMT) and demethylases (ten-eleven translocation demethylases or Tet) to run their enzymatic function over CpG sites in the vicinity, so hyper- and hypo-methylated gene landscape would be dynamically maintained upon cell needs [13]. There are two main functions of DNMTs. The first one is linked to preserving DNA methylation status in dividing cells driven mainly by DNMT1 over hemimethylated CpG sites [13]. Additionally, a second role is associated with *de novo* methylation during development by DNMT3A and DNMT3B anchored to nucleosomes [13]. However, these enzymes might help each other to maintain and newly synthesize methylated DNA [13].

DNA methylation and gene expression profiles are unique and specific to each cell type, including immune cell subsets [9]. Methylation profiling on peripheral blood mononuclear cells (PBMCs) samples may be masking cell-type-specific epigenetic signatures, such as marked differences in methylome studies between lymphoid and myeloid cells [11]. Therefore, methylation and transcriptomic studies carried on purified cell subsets will provide accurate results.

Methylome and transcriptome studies in female PBMCs demonstrated that monocytes and B cells display distinct and unique gene clustering while CD4^+^ and CD8^+^ T cells patterns gather together [11] (Figure 1). Differentially methylated genes in each immune cell subsets are located mainly downstream promoter-CGIs, exons, and introns [11]. However, several methylation differences could be found up or downstream shores and shelves [11]. Contrary to expected, the methylation status of promoter-bearing CpG sites displays fewer frequencies than total CpG sites [11]. Differential methylation genes (DMG) specific to cell subsets are not so frequent in CGIs. Additionally, the authors found that these non-CGIs’ differentially methylated regions (DMR) are located in enhancer elements and may regulate immune cell homeostasis functions [11]. These observations indicate that gene body methylation status plays a relevant role in cell-type-specific immune transcription and function.

### 1.2. Contribution of Demethylases to T and B Cell Differentiation

The DNA demethylation process requires a complex mechanism mainly driven by the ten-eleven translocation demethylases (Tet) [14]. This process includes the conversion of 5-methylcytosine (5mC) to 5-hydroxymethylcytosine (5hmC), 5-formylcytosine (5fC), and 5-carboxylcytosine (5caC). Then, these modified C are converted to unmodified C by base excision repair mechanisms using DNA glycosidase thymine-DNA glycosylase (TDG) [15]. An excellent work by Schoeleret al., using B cells derived from conditional knock-out mice, demonstrated that the lack of Tet2 and Tet3 impairs plasma cell differentiation without affecting cell proliferation [16]. CD138 expression and, IgG1 and IgE secretion after stimulation were markedly reduced in plasmablasts lacking Tet2/3 [16]. Furthermore, the immunization challenge showed that Tet2/3 deficiency limits the secretion of specific IgG1 while IgM was unaffected, suggesting that germinal centers maintenance and class switch recombination are dysfunctional [16]. In contrast, affinity maturation is not impaired in Tet2/3 conditional mice where new B cell clones numbers do not change [16].

Naive T cells usually displayed 5mC in transcriptional regulatory regions, such as promoter sites of cytokine locus overlapping with conserved non-coding sequences resulting in Th gene silencing (Figure 1). Non-coding DNA sequences may contain binding sites for transcription factors or other molecules involved in transcription regulation [17]. *IFNG*, *IL4*, and *IL17* genes displayed 5hmC modifications, specifically in conserved non-coding sequences (CNS) and promoter regions of purified Th1, Th2, and Th17 T cells [18]. The CNS6 enhancer sequence at the *IFNG* gene is most hydroxymethylated in Th1 cells and hypermethylated in the other Th subsets. Similarly, CNS2 and *IL17a* promoters of the *IL17* locus are highly hydroxymethylated in the Th17 subset but hypermethylated in different T cell subsets [18] (Figure 1). Thus, 5mC and 5hmC found at lineage-cytokine genes strictly link to their expression in each Th subsets and highlight that active DNA demethylation is crucial for immune regulation in Th-lineage development. Naive CD4^+^ T cells expressed high levels of Tet demethylases, but after TCR engagement, most Tet members are down-regulated [19]. However, Tet2 remains highly expressed in all Th subsets suggesting a broad role in Th differentiation. Furthermore, Th1 cells displayed recruitment of Tet2 to 5hmC-enriched CNS-6 and promoter regions of the *IFNG* locus where the presence of the T-bet transcription factor would be essential [19]. Similarly, Tet2 together with RORγt achieves DNA demethylation in the *IL17* locus. However, Th2-related genes are not as much targeted by Tet2 as Th1 or Th17. In contrast, as observed in CD4^+^ Th cells, during CD8^+^ T cells differentiation, Tet2 is much more linked to cell fate (effector/memory) than profile features or cytokine expression [18]. Tet2/3 are also associated with Pro-B to Pre-B transition or thymic T cell development as reviewed in Li et al. 2021; however, being outside the scope of the manuscript is not included [19].

## 2. Dysregulated Epigenetic Modifications in Autoimmune Diseases

### 2.1. Rheumatoid Arthritis (RA)

Rheumatoid arthritis (RA) is one of the most frequent chronic autoimmune diseases worldwide, with an annual incidence of 25–50/100,000 in Europe and the USA. This value has been steadily increased in recent years [20]. RA usually leads to joint destruction, disability, and premature death. It is well known that genetic factors are implicated in the development of arthritis and differ for the various forms of arthritis, with HLA class II (DR4) and HLA class I (B27) being associated with RA and spondyloarthritis (SpA), respectively. The mentioned genetic predisposition combined with environmental and epigenetic factors (which can be heritable) are involved in the development and chronicity of RA disease. Interestingly, some studies have evaluated the development of RA in identically genetic monozygotic twins to differentiate the effect of environmental factors on the predisposition to develop autoimmunity [21]. Thus, it has been reported a higher RA concordance rate (9.3–15.6%) in monozygotic twins than in dizygotic twins (2.3–3.6%) [22,23,24]. Interestingly, whether certain epigenetic modifications associated with RA are stable enough to be heritable is still unknown.

An increasing body of evidence suggests an important role for epigenetic alterations in the regulation of RA pathogenesis. The epigenetic modifications in synovial fibroblasts from RA patients have been of particular interest because of their known aggressive phenotype that remains stable for several passages in cell culture. Thus, synovial fibroblasts from RA patients are intrinsically activated by DNA hypomethylation, inducing gene upregulation [25]. In addition, several research groups have described alterations in the DNA methylome from fibroblasts-like synoviocytes in RA patients (Figure 2 and Table 1) [25,26,27,28].

On the other hand, some epigenetic changes can be considered as possible inflammation markers. For example, the expression of methyl CpG-binding domain 2 (MBD2) and DNMT1 are significantly augmented in RA patients [40]. However, these are not RA-specific alterations, and SLE patients share these characteristics [40].

Alterations in DNA methylation are observed in the MHC region of CD14^+^ monocytes from RA patients, associated with altered gene regulation and an increased RA risk [29]. Besides, other authors have reported hypermethylation of *IL10* [41] and *IL6* [42] promoter regions in PBMCs from RA patients. Additionally, hypomethylation of the promoter of *IL6* and *ERα* has been associated with increased expression in RA patients [43,44]. These alterations may be linked to increased IL-6 expression and Th17 cell proliferation [42]. Several hypomethylated genes have been detected in RA patients, such as *GALNT9* in B and T cells [45]. Moreover, *IL-6R*, *CAPN8*, *DPP4*, *CD74*, *CCR6*, and several *HOX* genes in fibroblast-like synoviocytes have DNA methylation changes and accordingly showed a dysregulated expression (Figure 2) [28]. Interestingly, there are different interactions between DNA methylations and miRNAs affecting gene regulation in an integrated way in RA patients [28].

Accordingly, miRNA have also been suggested as critical players in RA development. Thus, miR-146a abundance is associated with IL-17 expression in PBMC and RA synovium [46] and is reduced in Treg after stimulation [47]. On the other hand, some miRNA can regulate PTM; thus, an increased level of miR-126 reduces the *DNMT1* expression leading to hypomethylation of specific genes [48].

An evident sex bias is observed in the incidence and the course of RA, being more frequently diagnosed in women and developing a more aggressive disease [49], but its mechanisms are mostly unknown. Although a possible role of sex hormones in this predisposition has been indicated, it has also been suggested that epigenetic mechanisms related to sex chromosomes are also implicated. A recent study reports 81 methylated biomarkers in both regulatory regions and the gene body. Interestingly, only 38 markers were present in the Y chromosome, indicating the sex-based differences observed in RA patients [50].

Recently, the epigenetic landscape was evaluated in RA fibroblast-like synoviocytes, which adopt an aggressive phenotype in RA patients. They studied histone modifications, chromatin structure, RNA expression, and DNA methylation and detected epigenetic changes associated with active enhancers, promoters, and specific transcription factor binding motifs [30]. Interestingly, this study reports a new way to identify unexpected RA-specific targets relevant to the development of novel therapeutic agents by considering the complexity of the epigenomic landscape [30].

On the other hand, PTM alterations in histones have also been described in RA patients. Thus, the increased expression of histone deacetylase (HDCA) in PBMCs from RA patients compared to healthy individuals led to the application of HDCA inhibitors with beneficial effects reported on RA development [51], despite the side effects associated with non-selective HDAC inhibitors [52]. Besides, the importance of the HDCA1 enzyme in arthritis is highlighted by the study of T cell specific *HDCA1* KO mice. These mice are resistant to developing collagen-induced arthritis (CIA), although they produce anti-collagen antibodies, indicating a critical role of HDC1 in the T cell-dependent response in autoimmunity [53]. Furthermore, in synovial fibroblasts, increased HDAC expression and activity have also been detected [54]. Accordingly, the use of selective HDAC3 inhibitors has been reported as a potentially beneficial therapy for inflammatory disorders, including RA [55]. Moreover, the beneficial effect of an HDAC6 inhibitor has been observed by suppressing inflammatory responses on monocytes/macrophages [56].

Because epigenetic events are theoretically reversible, epigenetic intervention has significant therapeutic potential. In this context, the use of HDAC inhibitors has shown excellent anti-inflammatory effects in vitro and in animal models of RA [55,57]. Recently, another HDAC has been evaluated in the rat RA model, showing significant clinical score improvement, mobility, and inflammation reduction [58]. A similar outcome was reported in RA patients administered orally for three months with an HDAC inhibitor. These patients showed improved mobility, reduced number of swollen joints, and pain [59].

### 2.2. Systemic Lupus Erythematosus (SLE)

SLE is mainly driven by B cells; however, several reports also link lupus immunopathogenesis with T cells, Dendritic cells and monocytes [60,61,62,63]. Furthermore, DNA methylation studies have also linked T cells to lupus pathogenesis [64,65]. Modifying DNA methylation of T cells during polyclonal proliferation with the DNMT inhibitors, 5-azacitidine (5Aza) and procainamide, may drive aberrant pro-inflammatory genes transcription and loss of tolerance [66]. The administration of activated and demethylated T cells into mice develops a Lupus-like disease, including anti-dsDNA production and glomerular immune complex deposition [66]. However, a different outcome was observed in a T cell-targeted 5Aza approach where demethylation occurs only in T cells [67]. MRL^lpr^ lupus mice treated with nanolipogels loaded with 5Aza and tagged with anti-CD4, or -CD8 monoclonal antibodies ameliorated skin rash, proteinuria glomerular damage is reduced, and the inflammatory infiltration is decreased [67]. Surprisingly, authors found that targeting CD4^+^ T cells with this nanolipogel loaded with 5Aza, Foxp3^+^ Tregs displayed a marked expansion in spleen cervical lymph nodes. The authors also showed that the 5Aza treatment favors Foxp3 expression by inhibiting methylation in humans and mice treated CD4^+^ T cells. When nanolipogels were directed against CD8^+^ T cells, double-negative T cell subsets were reduced highly, suggesting a link between these two T cell populations [67]. In experimental models, absolute numbers of T and B cells, plasma cells, germinal center B cells, IFN-γ producing T cells, and effector/memory T cells were increased in the absence of Tet2 and Tet3 demethylases on B cells [68]. Furthermore, Tet2 and Tet3 deficiency leads to anti-dsDNA, -histone, and sm/RNP autoantibodies development, leading to renal immune-complexes deposition, which are significant features of lupus-like symptoms [68]. Indeed, when authors depleted CD4^+^ T cells or deleting H2-Ab1 (MHCII) gene, which prevents T-B cooperation, plasma cell numbers and T cell aberrant activation were decreased, suggesting that lymphocyte interaction is crucial in autoimmune initiation. In this work, the authors highlight the role of CD86 dysregulation on B cells and the subsequent T cell aberrant activation upon Tet2 and Tet3 deficiency in lupus-like disease [68]. Remarkably, the authors concluded that the lack of Tet2 and Tet3 conditions unleashes CD86 expression during continuous self-antigen exposure [68].

Inherited risk genetic including genome and epigenome does not lead to lupus development by itself, so environmental agents may apply [69]. UV light, procainamide, and hydralazine promoted lupus activation in several experimental models and was linked to human lupus flares [70] (Figure 3A and Table 1). Now, clarifying the picture, there is evidence that UV light, procainamide, and hydralazine are DNA methylation inhibitors that may lead to inadequate gene expression of immune cells, tolerance loss, and autoimmunity in susceptible hosts with genetic risk [71]. Large amounts of studies support the notion that lupus patients display an exacerbated DNA demethylated pattern. However, understanding the clinical role of these demethylation patterns in the SLE disease activity index (SLEDAI) score or even during lupus nephritis remains unclear. Identifying DNA methylation sites linked to disease activity and specific manifestations will provide new tools for executing precision medicine protocols in lupus.

In a cohort of SLE patients, 4839 and 1568 methylation sites were identified that were negatively and positively correlated with active disease. Interestingly, negatively correlated genes were enriched on chromosomes 3, 17, and 1, while positively correlated genes belong mainly to chromosome X [31] (Figure 3B). In this report, gene methylation positively and negatively associated with SLEDAI displayed a differential distribution primary on the nearest promoter region (less than 1 kb). These authors demonstrated that lupus patients decreased methylation status in crucial Th cytokines, such as *IL4*, *IL5*, *IL9*, *IL13*, *IL12B*, *IL17F*, and *IL22*, which correlates with active disease. *RORγt* and *BCL-6* genes were hypomethylated during active disease, while *T-bet* and *GATA-3* displayed a hypermethylated status [31]. However, RNA sequencing assays demonstrated that most DNA hypomethylation or hypermethylation genes positively or negatively correlated with disease activity from lupus patients displayed no changes in RNA expression linked to disease activity [31].

Additionally, DNA methylation arrays comparing African American vs. European lupus cohorts demonstrated that lupus patients display a methylated landscape that is very stable over time and linked to disease activity [32]. Two main loci, *SNX18* and *GALNT18*, were associated with disease activity and active lupus nephritis [32]. Additionally, as expected, the IFN signature associated genes, *STAT4*, and *NF-κB* signaling genes display differential methylation status [32]. Although IFN-related genes link mainly to SLE pathogenesis, several autoimmune diseases show an altered IFN gene expression. DNA methylation profiling of CD4^+^ T cells from Grave’s Disease (GD), RA, SLE, and Systemic Sclerosis (SSc) patients share a predominant hypomethylation pattern [33]. Strikingly, many type I-IFN-related genes display decreased methylation levels sharing a common hypomethylation pattern in GD, RA, SLE, and SSc [33]. Furthermore, these type I IFN-related genes exhibit good performance as diagnostics biomarkers of these autoimmune diseases. Similarly, *IFI44L*, a leading IFN signature gene, shows aberrant DNA methylation in CD4^+^ T cells from GD, RA, SLE, and SSc [33]. These data underscore the importance of research leading to various shared genes between different autoimmune diseases for their correct diagnosis and follow-up.

### 2.3. Sjogren’s Syndrome (SS)

This chronic autoimmune disorder has a higher female predisposition, similar to SLE. It is typically characterized by lymphocytic infiltration of salivary and lacrimal glands causing a reduced function [34]. SS can be classified as primary or secondary and shares high comorbidity with SLE and RA [72]. Moreover, SS is confirmed by the presence of anti-double-stranded DNA antibodies, Anti-Ro (anti-SSa), and Anti-La (anti-SSb) [73]. However, the lack of particular SS biomarkers has presented a challenge in the precise diagnosis of SS. A DNA methylation landscape was shown for SS and SLE, demonstrating that SS presents hypomethylation levels than healthy controls, such as type I interferon-induced genes. By comparing with SLE, SS patients display an increased methylation level [34]. Additionally, they identify differential methylation sites for primary SS, such as hypomethylation at the MHC class II locus *HLA-DPA1* (cg25824217) (Table 1) [34].

The reduced DNA methylation levels are also found in salivary gland epithelial cells; these hypomethylation levels are correlated with greater severity and B cells infiltration. However, since epigenetic changes are dynamics, administration of anti-CD20 monoclonal antibody rituximab as therapy for SS has shown an increment in DNA methylation levels [74]. Furthermore, hypomethylations in SS have been related to upregulation of costimulatory genes, such as *CD70* in CD4^+^ T cells promoting plasma cell differentiation and IgG production; pro-inflammatory cytokines, such as IFN-regulated genes, which is consistent with the IFN hallmark observed in SS patients [35]. However, other genes, such as *FOXP3* are hypermethylated, triggering a reduced regulatory T cell population and unbalanced immune response [75].

### 2.4. T Cell-Mediated Diseases: Multiple Sclerosis and Psoriasis

#### 2.4.1. Multiple Sclerosis (MS)

The presentation of MS symptoms is classified in two phases, including relapsing-remitting form (RR-MS) characterized by episodes of relapse and periods of clinical remission, and secondary-progressive (SP-MS), which causes more disability [76]. Studies of DNA methylation changes have shown that lymphocytes and monocytes from patients with RR-MS present a hypermethylation profile compared to healthy controls, which can be correlated with inflammation and clinical activity of MS since treatment with IFNβ significantly reduce the methylation profile [77]. However, a much more detailed study regarding DNA methylation elicits that the hypermethylation found in RR-MS patients when limited to CD4^+^ T cells can be correlated with MIR21 methylation. This gene is localized in the locus associated with MS susceptibility. Consequently, RR-MS displayed lower levels of miR-21 compared to SP-MS and healthy controls, suggesting a future target for therapies or used as an epigenetic biomarker [36]. Similarly, hypomethylation at vitamin D-receptor genes has been proposed as MS risk genes (Figure 4 and Table 1) [37]. In addition, the differential susceptibility to environmental stimuli during the first five years of life and how these changes persist into adulthood while the stimuli also persist were recently described [37].

#### 2.4.2. DNA Methylation in Psoriasis

Psoriasis is linked to aberrant crosstalk between dendritic cells, T cells, and keratinocytes to produce multiple inflammatory cytokines and growth factors [78,79,80,81,82]. Strikingly, although extensive studies over immune skin cells have been done, regulation of the immune response by DNA methylation in psoriasis has barely been addressed. Interestingly, purified blood CD4^+^ cells from discordant monozygotic twins, one healthy and one affected with psoriasis, display a highly similar DNA methylation landscape [38]. However, minor differences in DNA methylation and expression exist in CD4^+^ T cells that gather genes, such as *IL13* and *TNFSF11*, among others [38]. In addition, gene methylation profiles of psoriasis patients indicate that genes belonging to the IL17 signaling pathway, *Staphylococcus aureus* infection, interferons, and immune cells migration displayed abnormal methylation, including *IRF7*, *IL7R*, and *CXCL1* [83]. Interestingly, the imiquimod induced psoriasis model in knocking down Tet2 mice displayed decreased skin lesions with a reduced expression of biomarkers genes, such as *S100A7*, *IL7R*, and *IRF7*. These data suggest that deficient methylation/demethylation homeostasis may contribute to disease risk.

Psoriasis risk has been strongly associated with some HLA alleles, the skin DNA methylome of HLA-Cw*0602 bearing patients [39]. Interestingly, the authors showed that more than 500 and 2000 CpG sites were hypo- and hyper-methylated, respectively (˃10% methylation difference) (Figure 4) [39]. Furthermore, these DMSs for hypo- and hypermethylated sites locate in different CpG regions with more prevalence (≈50%) in CGIs, followed by open sea regions (≈25%), shores (≈20%), and shelf (≈6%). However, the authors could not find a clear association between the most significant DMSs and their gene expression in this study, probably due to the whole skin sample instead of purified cell origin [39]. Nevertheless, these data highlight the worth of continued work in autoimmune methylomes.

## 3. Female and X-Linked DNA Methylation

The higher prevalence of autoimmunity events in females than males might be due to sex-linked hormones and sex-associated methylation of X and autosomal chromosomes events between others [84,85]. Interestingly, it was shown that purified monocytes, B cells, CD4^+^ and CD8^+^ T cells from PBMCs displayed differences ranging 77 to 90% in methylation status between females and males [11]. The methylation landscape is very complex, where each cell-type-specific methylome may display both specific hypo- and hyper-methylated CpGs profiles in non-promoter or CGIs. Similarly, the same authors showed that differential methylated genes linked to the sex signature in autosomal chromosomes are mainly found in CGIs [11]. Indeed, sex-specific methylation could be edited by sex-endocrine factors, such as DNMTs induction during gestation [86]. It is proposed that sex-linked differential methylated regions initiate during development and strengthen by hormones during puberty. It has been reported that estrogen receptor (ER)α display noticeable inflammatory properties supporting disease development reflected in renal damage, using experimental lupus models [87,88,89]. Studies often show that ERα may be linked to DNA binding to modulate immune cell functions, as suggested by Cunningham et al., which demonstrated TLR-induced immune response requires direct binding to estrogen response elements [90].

In contrast, ERβ may display anti-inflammatory functions [88]. Interestingly, lower levels of ERβ may be found in lupus T cells [91]. Similarly, Crohn’s disease patients may also display a decrease in ERβ expression in blood T cells [92]. Although these data suggest that downregulation of ERβ may be linked to a pro-inflammatory condition, the correlation between expression and methylation has not yet been studied.

Interestingly, Golden et al. demonstrated in in-vitro assays that T cells displayed more methylation in CGIs of chromosome X when this was inherited from the father than the mother [85]. Additionally, the authors showed that offspring displayed preferred gene expression when inherited from a maternal X origin [85]. Notably, in this work, the authors highlight the *TLR7* gene, located in the X chromosome and is involved in lupus pathogenesis, such as observed in the lupus-mice Yaa [93]. Golden et al. demonstrated that *TLR7* is much more expressed when is the X chromosome comes from the father than the mother reinforcing the concept of epigenetic control of transcription and their associations with sex bias [85]. Similarly, Souyriset al. demonstrate that *TLR7* transcription on B and myeloid cells may often occur in both X chromosomes from healthy women and Klinefelter’s syndrome men, which may also be linked to increased disease risk in these men [94]. Additionally, lupus flares have also been proposed to be linked to methylation status. Swalhaet al. reported that combining the total genetic risk with the demethylation status of two T cell related loci linked to lupus, *KIR2DL4,* and *PRF1*, men may need much more DNA demethylation to achieve similar lupus flares than women. Similarly, genetic load and demethylation status in T cells correlate strongly with disease severity [95].

Biological markers are used to determine a normal situation, pathological processes, or the result of therapeutic interventions [96]. Thus, determining molecular markers in target tissue within the context of autoimmune diseases allows an in-depth understanding of the pathogenesis and the identification of new early diagnosis points and possible novel therapeutic targets. In addition, immune-related genes and inflammation pathways are under epigenetic-mediated regulation [4]. Accordingly, understanding the involvement of DNA methylation and histones modification in the pathogenesis of autoimmunity could provide a patient-specific drug-response prediction.

The phenotype heterogeneity and overlap within autoimmune diseases are some of the main aspects that make diagnosis difficult. However, early treatment of the disease can delay the onset of detrimental symptoms [97]. Therefore, this makes prompt intervention and accurate diagnosis critical for the patient’s progression. Although we have detailed multiple epigenetic alterations associated with autoimmune diseases throughout this review, most of these hallmarks have not yet been studied as biomarkers that allow early diagnosis. In addition, some epigenetic changes have been associated with disease progression. For example, low methylation of *CYP2E1* and *DUSP22* promoters have been associated with disease activity and could be used as a RA disease activity biomarker [98].

Similarly, hypomethylation of the promoter region of IFN-induced protein 44-like has been identified as a biomarker for SLE diagnosis with high sensitivity and specificity [99]. Interestingly, higher methylation is observed in SLE patients during remission, allowing the evaluation of the disease activity [99]. Besides, other renal-specific biomarkers suggested are *IRF7* [100] and carbohydrate sulfotransferase 12, which are hypomethylated in lupus nephritic patients [101]. On the other hand, in MS patients, disease- and state-specific changes have been reported to be linked to methylation patterns of cell-free plasma DNA, suggesting a potential biomarker for this disease [102]. Besides, H3 methylation (H3K9me2) and histone deacetylase (SIRT1) expression in PBMCs were reported as potential biomarkers for evaluating patients’ treatment responsiveness [103].

Plasma circulating miRNAs are ideal biomarkers for early autoimmune disease diagnosis and monitoring progression because they are stable and non-invasively detected in fluids. Importantly, several miRNAs (MiR-24, miR-26a, and miR-125a-5p) were reported to be increased in plasma from RA patients and thus, have been suggested as possible non-invasive biomarkers [104]. Notably, miR-24 and miR-125a-5p increase were specific for RA disease, and its level was reduced in SLE and osteoarthritis patients [104]. Moreover, five miRNAs (miR-103a-3p, miR-155-5p, miR-200a-3p, miR-210-3p, and miR-146a-5p) were suggested as potential Type 1 Diabetes (T1D) biomarkers as they were dysregulated in recently-diagnosed T1D patients [105].

## 4. Advantages and Disadvantages of Epigenetic Therapy in Autoimmunity

There are alterations in the epigenetic landscape that are shared by different autoimmune diseases [33]. Nevertheless, the altered expression of particular genes may help diagnose and determine new therapies among specific autoimmune disorders. Along these lines, recent work reported that a group of autoimmune diseases (RA, SLE, GD and, SSc) share the hypomethylation of IFN-related genes in CD4^+^ T cells and could be used as a signature for various autoimmune disorders [33]. Accordingly, aberrant type I IFN function has been implicated in several of the mentioned autoimmune diseases [33]. On the other hand, as discussed in the article, many genes and enzymes have been targeted as potential therapies for autoimmune diseases. Currently, preclinical and clinical trials have been made to test their security and efficiency. For instance, inhibitors of HDAC are widely used in medicine, and ITF2357 (givinostat) is administrated to children with an anti-inflammatory purpose for treating systemic-onset juvenile idiopathic arthritis [57]. This HDAC inhibitor has also been tested in animal models of autoimmune diseases, such as RA with promising results [57]. However, it is essential to consider that there is still a lack of information regarding the contribution of epigenetics to immune and non-immune responses [75].

Another example is using HDAC6 inhibitor as a treatment for SLE and inflammatory bowel disease in rodent models, showing anti-inflammatory effects via CKD-506. Nevertheless, the mechanism regarding inflammatory and non-inflammatory response and the cells involved remains unknown [56].

On the other hand, targets already described may serve in some ethnic groups but not all. For example, in RA disease, the upregulation of miR-499 rs3746444 increased risk, particularly in Caucasians [106]. Therefore, studies must correlate target genes with different population characteristics regarding ethnic groups, sex, age, etc.

Comparing the epigenetic alterations found in the autoimmune conditions and described in different studies is a complex task. Each report evaluates diverse cells (synoviocytes, B and T cells, neutrophils, monocytes, etc.), even heterogeneous populations of cells, such as PBMCs, and different technologies for the DNA methylation determination (Table 1). On the other hand, both innate (trained immunity) and adaptive responses are regulated by epigenetic modifications. Therefore, they could exhibit altered methylation patterns that affect the development of autoimmunity. Even so, the hypermethylation of promoter regions of regulatory genes and/or the hypomethylation of inflammatory regions has been previously described [31,41,42].

Finally, it is crucial to consider that aside from epigenetics-targeted therapies, some regular treatment for autoimmune diseases may also lead to epigenetic profiles similar to healthy controls, such as methotrexate used for RA treatment. For example, it has been shown that methotrexate reduces methylation in the *FOXP3* gene, restoring the Treg function by increasing FoxP3 and CTLA4 expression [107], in contrast to anti-TNFα therapy which has not been associated with DNA hypomethylation restoration in RA patients [40]. Moreover, dietary changes and microbiota alterations lead to changes in epigenetic (local and systemically). Therefore, intervention strategies (pre and probiotics) could be suitable for modifying epigenetic alterations [108]. Besides, several environmental factors, such as smoking which reduces DNA methylation, could increase the epigenetic risk [109].

## 5. Conclusions

A vast number of studies link aberrant DNA methylation with autoimmunity, mainly hypomethylated modifications. However, the specific role of these demethylated profiles remains unclear. Particular hypomethylation of inflammatory and hypermethylation of suppressor elements may be responsible for the heterogeneity of autoimmunity. Identifying DNA methylation sites on specific immune cell subsets may shed light on understanding genetic risk and predict flares. Typical manifestations linked to specific differential methylated genes will provide new tools for executing precision medicine protocols for autoimmune diseases. These data highlight the need to understand the balance between DNA methylation/demethylation findings and their correlation with gene expression and disease activity.

## Figures and Tables

**Figure 1 ijms-22-11892-f001:**
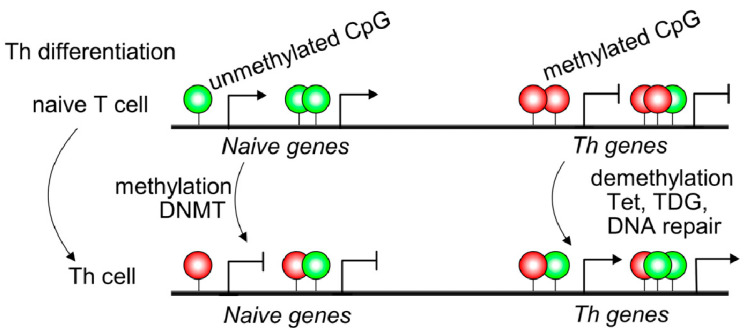
Methylation dynamics during Th differentiation. Naïve T cells display unmethylated and methylated cytosines on CCGG sites (CpG as green circles and mCpG as red circles), leading to gene expression (arrows) and repression (scale lines), respectively. During Th differentiation, dynamic control and remodeling on specific CpG sites occur. Genes linked to Naïve T cell biology may be repressed by several mechanisms, including DNA methylation on its CpG sites by methylases enzymes, such as DNA methyltransferases (DNMTs). By contrast, repressed Th genes on naïve T cells are induced under T cell priming conditions. These Th genes need to be demethylated by a group of enzymes which Tet2/3 are the most important converting methyl-C in hydroxymethyl-C. Then, this hydroxymethyl-C suffers serial reactions to result in C by DNA repair enzymes.

**Figure 2 ijms-22-11892-f002:**
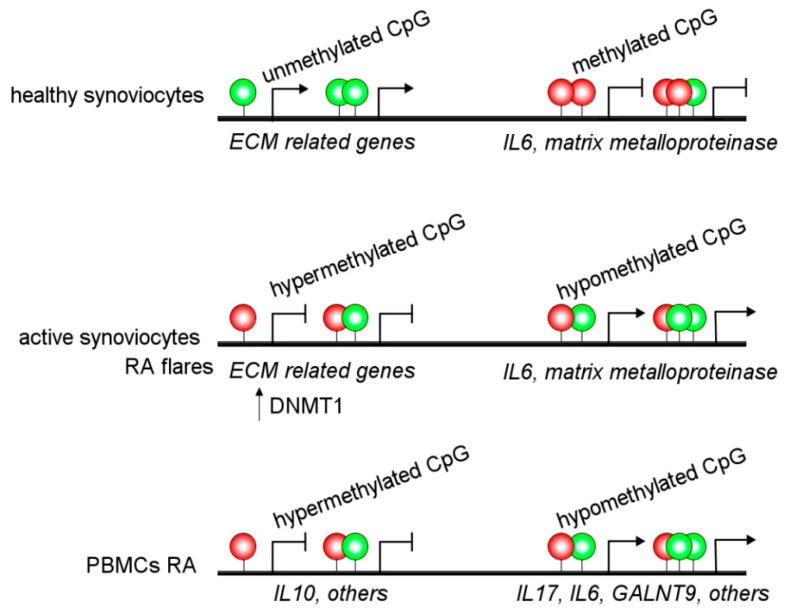
DNA methylation landscape in Rheumatoid Arthritis. Synoviocytes and peripheral blood mononuclear cells (PBMCs) from RA patients display an altered DNA methylation status, including hypomethylated (green circles) and hypermethylated (red circles) CpG sites. Both synoviocytes and PBMCs, display a predominant hypomethylated gene pattern compared to healthy controls. Gene expression of proinflammatory cytokines, such as *IL6* and *IL1* and extracellular matrix (ECM) degrading enzymes, such as metalloproteases were markers of active RA.

**Figure 3 ijms-22-11892-f003:**
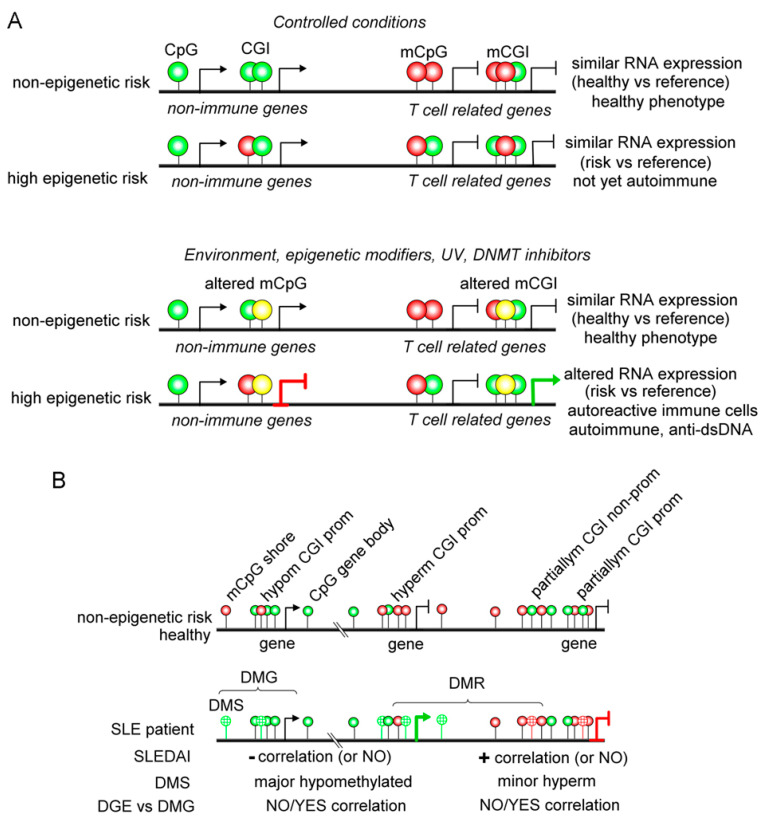
Hypothetical mechanism of DNA methylation in autoimmunity. (**A**) Autoimmune-susceptible hosts may carry on genetic risk and an altered DNA methylation status, including hypomethylated (green circles) and hypermethylated (red circles) CpG sites. However, this inheritable genetics is not sufficient to develop autoimmunity, as observed in monozygotic twins. Environmental agents, stress, UV, and epigenetic modifiers may alter methylated DNA status (yellow circles), leading to aberrant gene expression or repression. However, only high genetic-risk hosts may develop autoreactive immune cells resulting in autoimmune disease phenotype over time. (**B**) SLE methylome features on T cells. Lupus patients display differentially methylated regions (DMR) which are shaped by genes (differential methylation genes - DMG) and sites (DMS—grid circles). Most of the DMG/DMS seen in SLE display hypomethylated (green grid circles) patterns that may negatively correlate with SLE disease activity index (SLEDAI) and not often with gene expression. Hypermethylated (red grid circles) DMG/DMS is also seen in SLE patients that may positively associate with SLEDAI and not so often with gene expression. Genes could also be partially methylated (partiallym in the scheme).

**Figure 4 ijms-22-11892-f004:**
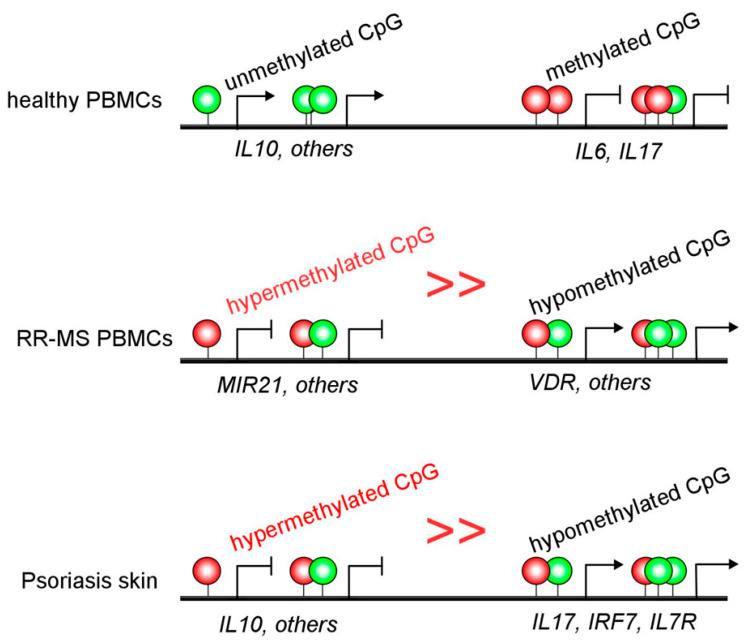
DNA methylation landscape in Relapsing Remitting-MS and Psoriasis autoimmunity. MS and Psoriasis patients carry an altered DNA methylation status, including hypomethylated (green circles) and hypermethylated (red circles) CpG sites. Most of the DMG/DMS seen in MS and Psoriasis display a hypermethylated pattern that may positively correlate with active disease and not often with gene expression. Hypomethylated (green circles) genes are also seen in both MS and Psoriasis. Although several genes, such as *IL17*, *IRFs*, *MIR*s, and *VDR*, have been proposed as epigenetics biomarkers for MS and Psoriasis, definitive validation is needed.

**Table 1 ijms-22-11892-t001:** Methylation studies in autoimmune diseases.

Condition	Methylation Modification (hypo/hyper)	Methods	Tissue/Cells	Disease Activity	Model/Population	Reference
	Global genomic hypomethylation. Fewer 5-methylcytosine and methylated CG sites upstream of an L1 open-reading frame	Immunohistochemistry for global 5-methylcytosine (5-MeC) determination and L1 promoter bisulfite sequencing	synovial fibroblasts from synovial tissue	Associated with activated phenotype in synovial fibroblasts	RA patients	[25]
	Hypomethylated loci in key genes (*CHI3L1*, *CASP1*, *STAT3*, *MAP3K5*, *MEFV* and *WISP3*). Hypermethylation in (*TGFBR2* and *FOXO1*)	Infinium HumanMethylation450 BeadChip. Methylation confirmed by pyrosequencing and gene expression by qPCR	fibroblast-like synoviocytes from synovial tissues	not mentioned	female RA patients	[26]
	1091 hypomethylated CpG sites (in 575 genes) and 1479 hypermethylated CpG sites (in 714 genes)	Integrated analysis of the DNA methylation, miRNA expression and mRNA expression data	fibroblast-like synoviocytes from synovial tissues	not mentioned	RA patients	[28]
	Two clusters within MHC regions with differential methylation potentially mediating genetic risk for RA	Illumina Human Hap300 v1.0 chip, Hap370CNVduo chip or Hap550duo chip	peripheral blood cells and monocyte cell fraction	not mentioned	RA patients with citrullinated protein antibodies, Swedish population	[29]
	No DNA methylation patterns identified but Huntingtin interacting protein-1 regulates FLS invasion into matrix	Histone modifications, WGBS, ATAC-seq and RNA-seq	synovial fibroblasts from synovial tissue	not mentioned	RA patients	[30]
SLE	4,839 hypomethylated and 1,568 hypermethylated CpG sites correlated	bisulfite genome-wide methylation assesment on Illumina platform. mRNA expression data	CD4+T cells	correlated negatively and positively with active disease	SLE patients, American	[31]
	487 hypomethylated and 420 hypermethylated CpG sites; *SNX18*, *GALNT18*, *IFN* signature genes	bisulfite genome-wide methylation assessment; Single nucleotide polymorphisms; Illumina platform	Neutrophils	correlated with Lupus nephritis	SLE patiens, African American and European American	[32]
	7889 hypomethylated and 7400 hypermethylated CpG sites; *IFI44L*	bisulfite genome-wide methylation assessment	CD4+ T cells	not mentioned	SLE, GD, RA and SSc	[33]
SS	509 Differentially methylated CpG sites, 5 unique for SS	EWAS with Illumina Human Methylation 450k Array	peripheral blood cells	Correlated with active disease	primary SS patients	[34]
	553 hypomethylated and 200 hypermethylated CpG sites	Genome wide DNA methylation with Illumina Human Methylation 450k Array	Naive CD4+ T cells	Correlated with changes in the pathogenesis of SS and with active disease	primary SS patients	[35]
MS	11 Hypermethylated CpG sites; *VMP1*, *MIR21*	Illumina Human Methylation 450k Array	CD4+ T cells	Correlated with Relapsing remitting MS	Relapsing remitting and secondary progressive form of MS patients	[36]
	502 Differentially methylated CpG sites	Bisulfite genome wide methylation assessment; Illumina platform; RADmeth software	CD14+ cells from haematopoietic progenitor cells	Correlation to incidence of MS and others autoimmune diseases	Adult and pediatric population	[37]
Psoriasis	*IL13*, *TNFSF11*, others	bisulfite genome-wide methylation assessment; Illumina platform	CD4+ and CD8+ T cells	not mentioned	Discordant Psoriasis twins’ patients	[38]
	811 hypomethylated and 3510 hypermethylated CpG sites; *IL17*, *IRF7*, *IL7*, *CXCL1*	bisulfite genome-wide methylation assessment; Genome-wide genotyping; Illumina platform	Skin samples	not mentioned	Psoriasis patients, HLA-Cw*0602 carriers	[39]

## Data Availability

Not applicable.

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
