# Peer review of "Contribution of Dysregulated DNA Methylation to Autoimmunity"

_ijms, 2021, doi:10.3390/ijms222111892_

Round 1

Reviewer 1 Report

Funes et al have piled a hundred research papers for this review. I appreciate this paper manifested in front of readers new progresses made recent years in the hot field – epigenetics in immunology. However, I have comments on how to efficaciously present these discoveries.

The Introduction gave an overall progress in epigenetics in immune system in the past. However, when the authors described each autoimmune disease, can more tables and Figures be created to guide readers?

For example, in 2.1 Rheumatoid arthritis (RA), with a Table, what was the first one looking into RA methylation, what method used, what was the discovery. List all representative studies.

If possible, depict all discoveries (for example the methylated genes involved in RA) into a Figure. This figure may help transform the extraction, comparison, summary, into a new hypothesis.

The same for other autoimmune diseases.

Finally, since this review focused on four autoimmune diseases, can the authors summarize what are the common and what are the difference among these four diseases in terms of discovery in DNA methylations.

Has this review included study of methylation gene enrichment analysis for these autoimmune diseases and show the differences among these autoimmune diseases? For example,

Chen S, Pu W, Guo S, Jin L, He D and Wang J (2019) Genome-Wide DNA Methylation Profiles Reveal Common Epigenetic Patterns of Interferon-Related Genes in Multiple Autoimmune Diseases. Front. Genet. 10:223. doi: 10.3389/fgene.2019.00223

Author Response

Answers to Reviewer 1:

  1. Reviewer 1: Funes et al have piled a hundred research papers for this review. I appreciate this paper manifested in front of readers new progresses made recent years in the hot field – epigenetics in immunology. However, I have comments on how to efficaciously present these discoveries.

Answer: We would like to thank you for the suggestions and effort to review this manuscript. We believe our manuscript was significantly improved after addressing the concerns and hope that the current revised version is acceptable for publication in the International Journal of Molecular Sciences.

  1. Reviewer 1: The Introduction gave an overall progress in epigenetics in immune system in the past. However, when the authors described each autoimmune disease, can more tables and Figures be created to guide readers?

Answer: As suggested by the Reviewer, we have included a new table (table 1) and figures (figures 2 and 4) to guide readers.

  1. Reviewer 1: For example, in 2.1 Rheumatoid arthritis (RA), with a Table, what was the first one looking into RA methylation, what method used, what was the discovery. List all representative studies.

Answer: As suggested by the Reviewer, we list the most representative studies looking for DNA methylation in RA in a table (table 1).

  1. Reviewer 1: If possible, depict all discoveries (for example the methylated genes involved in RA) into a Figure. This figure may help transform the extraction, comparison, summary, into a new hypothesis. The same for other autoimmune diseases.

Answer: As suggested by the Reviewer, we have created new figures with representative methylated genes involved in autoimmune diseases (figure 2 and 4).

  1. Reviewer 1: Finally, since this review focused on four autoimmune diseases, can the authors summarize what are the common and what are the difference among these four diseases in terms of discovery in DNA methylations.

Answer: As suggested by the Reviewer, we discussed differences and similarities between the mentioned autoimmune diseases (Page 12 Lines 410-415).

  1. Reviewer 1: Has this review included study of methylation gene enrichment analysis for these autoimmune diseases and show the differences among these autoimmune diseases? For example, Chen S, Pu W, Guo S, Jin L, He D and Wang J (2019) Genome-Wide DNA Methylation Profiles Reveal Common Epigenetic Patterns of Interferon-Related Genes in Multiple Autoimmune Diseases. Front. Genet. 10:223. doi: 10.3389/fgene.2019.00223

Answer: As suggested by the Reviewer, we have included and discussed the paper of Chen et al in the manuscript (Page 7 Lines 355-363 and Page 12 Lines 430-435).

We would like to thank again the Reviewers and the Editor for their time and effort to review this work. We hope that the current revised manuscript is acceptable for publication in the International Journal of Molecular Sciences.

Reviewer 2 Report

  This interesting review article has comprehensively discussed the epigenetic modifications, mainly DNA methylation, involved in autoimmune diseases in which T cells have a significant role such as rheumatoid arthritis and systemic lupus erythematosus. These disorders have differential gene methylation, mostly hypomethylated 5'-C-phosphate-G-3' (CpG) sites that may associate with clinical activity. The authors indicate that typical manifestations linked to specific differential methylated genes will provide new tools for executing precision medicine protocols for autoimmune diseases, and a better understanding of the impact of epigenetic modifications on the autoimmunity will contribute to the design of novel therapeutic approaches for these disorders.

  The manuscript is well written in English and the content is relevant to clinical application. There are only minor comments as follows.

  In the Abstract section lines 23, --- CpG sites ---. Instead of using abbreviation, the authors should display full name --- 5'-C-phosphate-G-3' (CpG) sites ---. In the Reference section line 493, the name of the journal should be Frontiers in immunology rather than J Frontiers in immunology.

Author Response

Answers to Reviewer 2:

  1. Reviewer 2: This interesting review article has comprehensively discussed the epigenetic modifications, mainly DNA methylation, involved in autoimmune diseases in which T cells have a significant role such as rheumatoid arthritis and systemic lupus erythematosus. These disorders have differential gene methylation, mostly hypomethylated 5'-C-phosphate-G-3' (CpG) sites that may associate with clinical activity. The authors indicate that typical manifestations linked to specific differential methylated genes will provide new tools for executing precision medicine protocols for autoimmune diseases, and a better understanding of the impact of epigenetic modifications on the autoimmunity will contribute to the design of novel therapeutic approaches for these disorders. The manuscript is well written in English and the content is relevant to clinical application. There are only minor comments as follows.

Answer: We would like to thank you for the revision made to this manuscript, and we also thank the esteem for our work. We believe our manuscript was significantly improved after addressing the concerns and hope that the current revised version is acceptable for publication in the International Journal of Molecular Sciences.

  1. Reviewer 2: In the Abstract section lines 23, --- CpG sites ---. Instead of using abbreviation, the authors should display full name --- 5'-C-phosphate-G-3' (CpG) sites ---. In the Reference section line 493, the name of the journal should be Frontiers in immunology rather than J Frontiers in immunology.

Answer: As suggested by the Reviewer, we have corrected the abbreviation errors (Page 1 Line 23).

We would like to thank again the Reviewers and the Editor for their time and effort to review this work. We hope that the current revised manuscript is acceptable for publication in the International Journal of Molecular Sciences.

Round 2

Reviewer 1 Report

The new version has been much improved with updated data and information. Thanks